# Detection of Feline Coronavirus Variants in Cats without Feline Infectious Peritonitis

**DOI:** 10.3390/v14081671

**Published:** 2022-07-29

**Authors:** Stéphanie Jähne, Sandra Felten, Michèle Bergmann, Katharina Erber, Kaspar Matiasek, Marina L. Meli, Regina Hofmann-Lehmann, Katrin Hartmann

**Affiliations:** 1Clinic of Small Animal Medicine, Centre for Clinical Veterinary Medicine, LMU Munich, Veterinärstraße 13, 80539 Munich, Germany; s.felten@medizinische-kleintierklinik.de (S.F.); n.bergmann@medizinische-kleintierklinik.de (M.B.); hartmann@uni-muenchen.de (K.H.); 2Section of Clinical and Comparative Neuropathology, Institute of Veterinary Pathology, Centre for Clinical Veterinary Medicine, LMU Munich, Veterinärstraße 13, 80539 Munich, Germany; erber@patho.vetmed.uni-muenchen.de (K.E.); kaspar.matiasek@neuropathologie.de (K.M.); 3Clinical Laboratory, Department of Clinical Diagnostics and Services, Center for Clinical Studies, Vetsuisse Faculty, University of Zurich, Winterthurerstrasse 260, 8057 Zurich, Switzerland; mmeli@vetclinics.uzh.ch (M.L.M.); rhofmann@vetclinics.uzh.ch (R.H.-L.)

**Keywords:** FCoV, FIP, RT-PCR, S gene mutation, sequencing, IHC, immunohistochemistry

## Abstract

(1) Background: This study aimed to detect feline coronavirus (FCoV) and characterize spike (S) gene mutation profiles in cats suffering from diseases other than feline infectious peritonitis (FIP) using commercial real-time reverse transcription polymerase chain reaction (RT-qPCR) and reevaluating results by sequencing. (2) Methods: In 87 cats in which FIP was excluded by histopathology and immunohistochemistry, FCoV 7b gene and S gene mutation RT-qPCR was performed prospectively on incisional biopsies and fine-needle aspirates of different organs, body fluids, and feces. Samples positive for S gene mutations or mixed FCoV underwent sequencing. (3) Results: In 21/87 cats, FCoV RNA was detectable. S gene mutations were detected by commercial RT-qPCR (and a diagnostic algorithm that was used at the time of sample submission) in at least one sample in 14/21 cats (66.7%), with only mutated FCoV in 2/21, only mixed in 1/21, and different results in 11/21 cats; in the remaining 7/21 cats, RNA load was too low to differentiate. However, sequencing of 8 tissue samples and 8 fecal samples of 9 cats did not confirm mutated FCoV in any of the FCoV RNA-positive cats without FIP. (4) Conclusions: Sequencing results did not confirm results of the commercial S gene mutation RT-qPCR.

## 1. Introduction

Feline coronavirus (FCoV) is the etiologic agent of feline infectious peritonitis (FIP). Worldwide, up to 90% of cats are infected with FCoV, especially those living in multi-cat environments, but only a small percentage of FCoV-infected cats (7–14% in multi-cat environments) develop FIP [1]. Cats with FCoV infection without FIP also can have viremia, and FCoV has been detected in organs of healthy cats [2,3,4,5].

There are two serotypes of FCoV: type I and type II [6,7,8]. Serotype I is considered to be of only feline origin, while serotype II originates from a recombination of canine coronavirus and feline coronavirus [8,9]. Both, type I and type II FCoV can cause FIP. FCoV infection is generally asymptomatic or leads to mild transient enteritis, while FIP is fatal if untreated. A key event towards the development of FIP is the development of mutations of the FCoV genome that are responsible for a change in viral cell tropism from enterocytes to monocytes/macrophages. FIP-associated FCoV have the ability to effectively replicate within monocytes/macrophages [10]. It is not exactly known which mutations are responsible for the switch in pathogenicity from less virulent FCoV to FIP-associated FCoV. None of the mutations that have been identified to date appear to consistently correlate with the FIP-associated pathotype [11,12,13,14]. However, as the FCoV spike (S) protein is considered responsible for receptor binding and viral cell entry [15], mutations in the FCoV S gene have been suggested to play an important role in the switch in cell tropism and pathogenicity [11,16,17,18]. By sequencing S genes of a large number of FCoV strains from feces of healthy cats and ascites or tissues of cats with FIP, two single nucleotide polymorphisms (SNP) in close proximity were found at nucleotide position 23531 and 23537 of the S gene only in FCoV of cats with FIP but not in FCoV of healthy cats. These SNPs at nucleotide position 23531 and 23537 lead to amino acid substitutions at position 1058 (M1058L) and 1060 (S1060A), respectively, within the spike protein. In one study, one of the SNPs was present in 96% of FCoV S genes in cats with FIP [11]. However, another research group [19] suggested that the SNPs in nucleotide 23531, resulting in amino acid substitution M1058L, would be more likely associated with any systemic spread of FCoV rather than with the FIP phenotype, since this substitution could also be detected in tissues from cats without FIP [19]. Therefore, the role of these S gene mutations in FIP pathogenesis is still unclear.

Diagnosis of FIP in practice is a challenge. The only confirmative method is immunohistochemical (IHC) staining of FCoV antigen in macrophages within tissues with characteristic changes [20,21,22], but this requires invasive tissue collection. Reverse transcription polymerase chain reaction (RT-PCR) is frequently applied to detect FCoV RNA in different samples, but FCoV RNA can also be detected in cats without FIP [15,19,20,21,22], although cats without FIP generally have significantly lower FCoV RNA loads measured by quantitative RT-PCR than cats with FIP [15,19,23,24,25,26]. Thus, quantification of FCoV RNA loads is an option to narrow the diagnosis of FIP, since the presence of high levels of FCoV RNA in tissue samples is highly suggestive of FIP (ABCD Diagnostic tree) [27].

Since a few years, a real-time RT-PCR (RT-qPCR) assay, aiming not only to detect all FCoV but specifically those with S gene mutations, is commercially available. A number of studies have investigated the value of this commercial assay in the diagnosis of FIP [25,28,29,30,31,32], but contrasting results were obtained and occasionally, FCoV with S gene mutations were detected using this assay in cats without FIP [25].

The aim of the present study was to evaluate how frequently different FCoV variants (those with the described S gene mutations and those without these mutations) occur in cats without FIP (definitively excluded by IHC) by analyzing many different tissues, body fluids, and feces, and thus, to characterize the S gene mutation profile of the FCoV found in cats without FIP. First, all samples were analyzed by a commercially available RT-qPCR detecting the presence of FCoV RNA (FCoV 7b gene RT-qPCR) followed in positive samples by an additional RT-qPCR for detection of presence or absence of S gene mutations (FCoV S gene mutation RT-qPCR). Second, the presence or absence of S gene mutations was reevaluated by sequencing of the S gene region.

## 2. Materials and Methods

### 2.1. Study Population and Exclusion of FIP

In total, 87 cats were prospectively included from July 2018 to August 2019. Inclusion criteria were (1) the definitive exclusion of FIP by necropsy, histopathology, and negative IHC of mesenteric and popliteal lymph nodes, liver, spleen, lung, omentum, kidney, intestine (duodenum, jejunum, ileum, colon), and brain, and (2) a definitive diagnosis of a disease other than FIP established by necropsy including histopathology (Appendix A).

For histopathology and IHC, specimens of all tissues were fixed by immersion in 10% buffered formalin, embedded in paraffin, and processed for histopathology and IHC. For histopathology, sections were stained by hematoxylin and eosin.

IHC was performed using an FIPV3-70 monoclonal antibody (Linaris Medizinische Produkte GmbH, Dossenheim, Germany) on formalin-fixed, paraffin-embedded tissue sections as described previously [33]. For signal detection, diaminobenzidine (DAB) staining was implemented (DAB Substrate Kit; Vector Laboratories, Burlingame, CA, USA). Negative controls were included in every immunohistochemical staining process. These were brain tissue samples from a cat with confirmed FIP affecting the central nervous system in which the antibody was substituted by phosphate buffered saline (PBS). Additionally, to ensure adequate performance of the antibody, a positive tissue control (tissue from a cat with confirmed FIP) was included in every IHC run. Samples were considered positive for FIP in IHC if typical histopathological lesions were present (e.g., pyogranulomatous and fibrinonecrotic tissue lesions at predilection sites with exclusion of other pathogens) and FCoV antigen was detected within macrophages in those lesions. Samples were considered negative for FIP in IHC if no histopathological lesions suggestive of FIP were detected and FCoV antigen was absent in all tissue samples including lymph nodes.

### 2.2. Sample Collection

Different samples (tissues, body fluids, and feces) (Table 2) were investigated for the presence of FCoV RNA. Tissue samples included fine-needle aspirates (FNA) as well as incisional biopsies of mesenteric and popliteal lymph nodes, liver, kidney, spleen, and lungs, and only incisional biopsies of omentum, intestines (duodenum, jejunum, ileum, colon), and brain.

For FNA, a 21-G needle and a 2 mL syringe were used. Cellular material was layered on two slides without staining. Incisional biopsies of all tissues were stored in Eppendorf tubes with 0.9% saline. Whole blood (ethylenediamine tetra acetic acid (EDTA) blood) was collected antemortem if possible; otherwise, blood was drawn postmortem by cardiac puncture. Additionally, body cavity fluid samples (effusion or peritoneal lavage samples), aqueous humor, and cerebrospinal fluid (CSF) were collected into Eppendorf tubes. In cats without effusion, peritoneal lavage was performed by instilling 20 mL/kg saline (0.9%) into the peritoneal cavity, massaging the abdomen, and withdrawing fluid by paracentesis. All samples were immediately stored at 4 °C until processing. Samples were sent to the commercial laboratory within 24 h of collection. Samples were mostly analyzed on the day of arrival at the commercial laboratory. If this was not possible, they were stored at 4 °C until analysis, but time between sampling and examination never exceeded 72 h.

### 2.3. FCoV 7b Gene RT-qPCR

First, FCoV 7b gene RT-qPCR was performed on all samples at a commercial laboratory (IDEXX Laboratories, Ludwigsburg, Germany). The RT-qPCR targeting the FCoV 7b gene was performed to detect the presence of FCoV RNA. Briefly, total nucleic acid was extracted from samples applying the “MagVet^TM^ Universal Kit” (ThermoFisher, Darmstadt, Germany) on a KingFisher Flex Purification System platform (ThermoFisher), and 7b gene RT-qPCR was performed using primers and probes as previously described [34].

### 2.4. FCoV S Gene Mutation RT-qPCR

In a second step, in all samples positive for FCoV RNA in the 7b gene RT-qPCR, two additional RT-qPCR assays targeting the S gene SNPs at nucleotide positions 23531 and 23537 (corresponding to the M1058L and S1060A substitutions within the fusion peptide of the S protein, respectively) were performed at the commercial laboratory (FCoV S gene mutation RT-qPCR; IDEXX Laboratories, Ludwigsburg, Germany). Briefly, highly specific hydrolysis probes were used to detect mutated sequences within the S gene. These probes were fluorophore-labeled (6-FAM and VIC, respectively). Results were analyzed to determine the 6-FAM:VIC (mutated FCoV: non-mutated FCoV) fluorescence ratio emitted by the hydrolysis probes. All sample types, including feces, were evaluated by the FCoV S gene mutation RT-qPCR. However, it should be mentioned that the commercial FCoV S gene mutation RT-qPCR has not been validated for use with feces, and fecal specimens are not an accepted sample type for this test by the commercial laboratory.

Interpretation of the results from FCoV S gene mutation RT-qPCR was performed by an algorithm. This algorithm (termed “original algorithm” in the following) used for interpretation was updated later (approximately September 2019) between the time the samples were collected (July 2018 to August 2019) and the evaluation of the data and preparation of the manuscript (“new algorithm” in the following). Originally, the “original algorithm” employed a subjective evaluation of the cycling curve for mutated FCoV to non-mutated FCoV fluorescence ratios that were <2. The presence of a strong curve was an indication for the presence of “mixed FCoV”.

Interpretation of the results using the current commercial algorithm (“new algorithm”) were as follows: (1) “negative” if no FCoV at all was detected by FCoV 7b gene RT-qPCR and no further S gene mutation RT-qPCR was performed, (2) “low” if FCoV load in the FCoV 7b gene RT-qPCR was below a cut-off of 1.5 million RNA equivalents per mL, and due to the low load, no further differentiation of the FCoV strains via FCoV S gene mutation RT-qPCR was possible, (3) “non-mutated FCoV” if FCoV was detected by FCoV 7b gene RT-qPCR but the mutated to non-mutated fluorescence ratio was <1.5, (4) “borderline” if a specific PCR product (amplicon) could neither be detected with certainty nor its presence could be ruled out, (5) “mutated FCoV” if the FCoV 7b gene RT-qPCR was positive and the mutated to non-mutated fluorescence ratio was >2 (either mutation in nucleotide 23531 or 23537), or (6) “mixed FCoV”, if the FCoV 7b gene RT-qPCR was positive and the fluorescence ratios of mutated to non-mutated fell in between “mutated FCoV” and “non-mutated FCoV” (1.5–2).

RT-qPCR was run with six quality controls, including: (1) RT-qPCR-positive controls (using synthetic DNA covering the real-time PCR target region); (2) RT-qPCR-negative controls (PCR-grade nuclease-free water); (3) negative extraction controls (extraction positions filled with lysis solution and PCR-grade nuclease free water only); (4) RNA pre-analytical quality control targeting feline ssr rRNA (18S rRNA) gene complex; (5) a swab-based environmental contamination monitoring control, and (6) an internal positive control spiked into the lysis solution to monitor the nucleic acid extraction efficiency, and presence or absence of inhibitory substances (using lambda phage DNA). These controls assessed the functionality of the PCR test protocols (1), for the absence of contamination in the reagents (2) and laboratory (5), absence of cross-contamination during the extraction process (3), quality and integrity of the RNA as a measure of sample quality (4), and absence of PCR inhibitory substances as a carry-over from the sample matrix (6).

### 2.5. Sanger Sequencing

Sanger sequencing was performed on all tissue and fluid samples (except feces) that were positive for a mutation either in nucleotide 23531 or 23537 in the FCoV S gene mutation RT-qPCR according to the original algorithm and on all fecal samples that were defined as “mixed FCoV” or “mutated FCoV” in the FCoV S gene mutation RT-qPCR according to the original algorithm.

All samples had been stored at −20 °C until processing. Retained samples were transferred to the Clinical Laboratory, Vetsuisse Faculty Zurich for RNA isolation and sequencing in April 2021. Viral RNA was isolated with the RNeasy Mini Kit (Qiagen, Hombrechtikon, Switzerland). Presence of viral RNA was re-assessed by quantitative 7b gene RT-PCR as described previously [34]. FCoV 7b gene RT-qPCR-positive samples underwent conventional RT-PCR amplifying part of the FCoV S gene containing nucleotides 23531 and 23537 [11,35]. RT-PCR amplicons were purified and sequenced either directly or after cloning into the pCR^®^II-TOPO plasmid with the TOPO TA Cloning Kit for Sequencing (Invitrogen, Thermo Scientific, Cleveland, OH, USA) as previously described [35,36]. For the amplification of the S gene targeting the single nucleotide polymorphisms leading to amino acid substitutions M1058L and S1060A, a modified previously published nested RT-PCR assay was used [11]. The FCoV-UCD1-S.3022 (5′-CAA TAT TAC AAT GGC ATA ATG G-3′) forward and FCoV-UCD1-S.3636 (5′-CCC TCG AGT CCC GCA GAA ACC ATA CCT A-3′) reverse primer were used for a first amplification (amplicon 615 bp) using the one-step RT-PCR kit (SuperScript III RT/Platinum Taq Mix, Invitrogen). The reaction had the same composition as above. Cycling conditions were 55 °C for 30 min, 94 °C for 2 min; 40 cycles of 94 °C for 15 s, 47 °C for 30 s, and 68 °C for 2 min; 5 min at 68 °C and then cooling to 10 °C. For the second nested PCR step, if needed, two newly designed primer pairs amplifying the region of interest (amplicon 134 bp), FCoV-UCD1-S.3027f (5’-AAT GGT GCT TCC TGG GGT TG-3’) and FCoV-UCD1-S.3160r (5’-GCA CCT GCA TAG CAA AAG GC-3’), and the Phusion Hot Start II High-Fidelity DNA Polymerase (Thermo Fisher Scientific, Waltham, MA, USA) were used. Five μL of one-step RT-PCR product were added to 20 μL of Mastermix (composed of 5 μL 5× HF Buffer, 0.5 μL d’NTPs (10 mM), 0.625 μL of each primer (20 μM), 0.5 μL Phusion High-Fidelity Polymerase (2 U/μL), and 12.75 μL RNase-free water). Cycling conditions were 98 °C for 3 min; 40 cycles of 98 °C for 10 s, 59 °C for 30 s, and 72 °C for 2 min; 10 min at 72 °C and then cooling to 10 °C. Sanger sequencing was performed by a commercial laboratory (Microsynth AG, Balgach, Switzerland) either using the RT-PCR primers or M13 forward and M13 reverse primers on 800 ng plasmid DNA as described previously [35]. Furthermore, this method had been applied on effusion and blood samples originating from histopathologically confirmed cases of FIP, where the mutated leucine at position 1058 was described, as reported by Meli et al. [36].

## 3. Results

### 3.1. Presence of FCoV RNA in Cats without FIP

A total of 1861 samples of 87 cats were analyzed by FCoV 7b gene RT-qPCR for presence of FCoV RNA. FCoV RNA in at least one sample was detected in 21 of the 87 cats (24.1%). A total of 78 samples of these 21 cats were FCoV 7b gene RT-qPCR-positive (Table 1 and Table 2).

### 3.2. Results of FCoV S Gene Mutation RT-qPCR following the Original Algorithm

FCoV S gene mutation RT-qPCR detected “mutated FCoV” (with a fluorescence ratio of mutated to non-mutated FCoV > 2) and “mixed FCoV” in at least one sample in 14 of the 21 FCoV 7b gene RT-qPCR-positive cats (66.7%; only “mutated FCoV” were detected in 2/21 (9.5%), only “mixed FCoV” in 1/21 (4.8%), and different results with at least one sample positive for “mixed FCoV” or “mutated FCoV” in different tissues were found in 11/21 (52.4%) cats). Based on the original algorithm, none of the positive samples were interpreted as “non-mutated FCoV”. The remaining 7/21 (33.3%) cats were FCoV 7b gene RT-qPCR-positive but RNA load was too low in all tested samples to differentiate between mutated and non-mutated FCoV (“low”) (Table 1 and Table 2, Figure 1). The crossing point (cq) values of the positive samples are shown in Figure 2.

### 3.3. Results of FCoV S Gene Mutation RT-qPCR following the New Algorithm

FCoV S gene mutation RT-qPCR detected “mutated FCoV” (with a fluorescence ratio of mutated to non-mutated FCoV > 2) and “mixed FCoV” in at least one sample in 10 of the 21 FCoV 7b gene RT-qPCR-positive cats (47.6%; only “mutated FCoV” were detected in 2/21 (9.5%), only “mixed FCoV” in 0/21 (0%), and different results with at least one sample positive for “mixed FCoV” or “mutated FCoV” in different tissues were found in 8/21 (38.1%) cats). Different results with samples positive for “non-mutated FCoV” and “low” were detected in 4/21 cats (19.0%). The remaining 7/21 (33.3%) cats were FCoV 7b gene RT-qPCR-positive but RNA load was too low in all tested samples to differentiate between mutated and non-mutated FCoV (“low”) (Table 1 and Table 2, Figure 1).

### 3.4. Results of Sanger Sequencing

In total, 24 samples (14 body fluid and tissue samples and 10 fecal samples) of the 87 samples identified by the original algorithm as “FCoV with S gene mutations” or as “mixed FCoV” were selected for sequencing. Sequencing was possible in 18/24 samples (9/14 body fluid and tissue samples and 9/10 fecal samples) with 16/18 successfully sequenced samples (8 tissue and 8 fecal samples). This sample set included 3 fecal specimens and 1 sample from the ileum that were subsequently recategorized as “non-mutated FCoV” based upon the new algorithm. Only non-mutated FCoV was detected in all 16 samples (Table 3) by direct sequencing of the RT-PCR amplicons.

Of these 16 samples, 4 tissue samples and 4 fecal samples were selected for further characterization by cloning of the RT-PCR amplicons and sequencing of 10 clones each to detect possible underrepresented sequence variants. In all sequenced clones, only non-mutated FCoV was detected.

## 4. Discussion

This is the first study that included a large number of different tissue, body fluid, and fecal samples from a defined population of cats in which FIP was definitively ruled out by gold standard methods. The results of the commercially available FCoV S gene mutation RT-qPCR were quite extraordinary and unexpected because mutated FCoV was found in many tissues of these non-FIP cats and even in fecal samples. Therefore, reevaluation by Sanger sequencing was performed. This also allowed discussion about specificity of the commercially available FCoV S gene mutation RT-qPCR in the diagnosis of FIP.

The commercially available FCoV S gene mutation RT-qPCR used in the present study was developed to detect mutations considered specific for FIP and therefore, to have a higher diagnostic specificity to diagnose FIP than the detection of all FCoV by FCoV 7b gene RT-qPCR. Interestingly, the commercial assay diagnosed positive FCoV S gene mutation RT-qPCR results in several tissue, fluid, and even fecal samples of these cats in which FIP was definitively ruled out. However, it should be noted that fecal samples are not validated for use with this FCoV S gene mutation RT-qPCR by the commercial laboratory. Still, the fact that mutated FCoV was also found in feces using the FCoV S gene mutation RT-qPCR and that according to the original algorithm, non-mutated FCoV was not at all found in any sample of any of the tested cats, not even in feces, was very unexpected. The fluorescence from both, the mutated and non-mutated probes was detected in the FCoV S gene mutation RT-qPCR; however, using the original algorithm based on the ratio and the cycling curve, none of the results were interpreted as “non-mutated”. To further clarify the presence of S gene mutations in cats without FIP, samples positive by FCoV S gene mutation RT-qPCR were subsequently analyzed by Sanger sequencing of the S gene region of interest. Strikingly, sequencing did not confirm the results of the FCoV S gene mutation RT-qPCR in any of the sequenced samples; only non-mutated FCoV could be detected by sequencing. These contrasting results were highly surprising. Possible explanations include methodological errors of one of the diagnostic methods. Erroneous results of the Sanger sequencing seem highly unlikely, since the validity of the sequencing assay used in the present study has been confirmed previously [36] using 15 samples of 15 cats with confirmed FIP (and positive FCoV S gene mutation RT-qPCR): in 6 cats, the S gene mutation leading to amino acid substitution M1058L could be confirmed by sequencing; in 2 cats, the S gene mutation leading to amino acid substitution S1060A was present, and in 1 cat, a S gene mutation leading to amino acid substitution M1058F was present; in 4 cats, sequencing could not be performed because of low viral load [36]. Furthermore, to exclude the presence of a small proportion of mutated together with non-mutated FCoV (mixed FCoV) in the present study, the RT-PCR amplicon was cloned into a plasmid in selected samples, and 10 clones were sequenced from each amplicon. Even with this approach, no mutated FCoV could be detected in any of the analyzed samples. While these results do not completely rule out a very low number of mutated FCoV besides non-mutated FCoV, they do call the diagnostic value of the FCoV S gene mutation RT-qPCR into question.

So far, most previous studies evaluating the diagnostic specificity of the commercial FCoV S gene mutation RT-qPCR were hampered by the fact that they often did not include control cats without FIP that were positive for FCoV RNA by FCoV 7b gene RT-qPCR. Thus, a clear statement about the diagnostic specificity of the FCoV S gene mutation RT-qPCR could not be made [30,31]. Additionally, previous studies only included a small number of cats without FIP [25,28,30]. Still, one study also reported a positive FCoV S gene mutation RT-qPCR result in one cat in which FIP was definitely excluded [25]; mutated FCoV (amino acid substitution M1058L) was found in an effusion sample in this cat that suffered from chronic kidney disease. In those previous studies, it was discussed that positive results of the commercial FCoV S gene mutation RT-qPCR in cats without FIP could potentially be explained by the fact that cats with positive results could have suffered from early-stage FIP, although there were no histopathological changes of FIP. Furthermore, mutations in the S gene’s nucleotides 23531 and 23537 leading to amino acid substitutions M1058L and S1060A have been discussed as being markers of systemic spread of FCoV rather than being markers of the FIP phenotype [19]. Finally, a true false positive result because of a methodological error was possible. The fact that in the present study, sequencing (whenever it was possible) only detected FCoV with S genes without mutations in all the cats without FIP makes misdiagnosis and early-stage FIP as an explanation very unlikely.

Additionally, these findings indicate that the S gene mutations, if detected by sequencing, could indeed be specific for FIP, which was suggested by earlier studies [11,37], and that sequencing can be used to distinguish between cats with FIP and cats without FIP. This is in contrast to studies, in which presence of M1058L or S1060A substitutions in the FCoV S protein were supposed to be a hallmark of systemic spread of FCoV rather than a definitive marker of FIP [19,23]. In these studies, mutated FCoV was detected by sequencing in cats without FIP in tissue and effusion samples [23]. However, the results of the present study, in which mutated FCoV could not be detected in tissues from cats without FIP, together with the results of the study by Meli et al. [36] in which mutated FCoV was indeed detected by sequencing in various tissues and body fluids of cats with FIP, support the hypothesis that the mutations are indeed a marker for FIP and not only for systemic spread of FCoV [19].

As determined by Sanger sequencing, non-mutated FCoV was found in all 9 cats without FIP of which samples could successfully be sequenced. Three of the 9 cats had only a single sample sequenced and that sample was recategorized by the new algorithm to be non-mutated. In these cats, non-mutated FCoV could not only be detected in the feces, but also in various tissues. The presence of non-mutated FCoV in feces and in the colon and ileum was to be expected, as enteric infection and viral persistence has been described in otherwise clinically healthy cats and in cats suffering from diseases other than FIP [2,3,38]. Interestingly, however, non-mutated FCoV was also detected in other tissues, such as mesenteric and popliteal lymph nodes and kidneys. This has been shown before [35] and underlines the fact that non-mutated FCoV can spread systemically not only in cats with, but also in cats without FIP, and that the detection of FCoV by FCoV 7b gene RT-qPCR per se is not specific for FIP. Thus, a positive FCoV 7b gene RT-qPCR result, particularly if without determination of the viral load, cannot be used as a single diagnostic test to diagnose FIP.

FIP is a fatal disease if left untreated, and thus, specificity of confirmative tests is essential to avoid euthanasia or potentially harmful and expensive treatments in cats misdiagnosed with FIP. The results of the present study indicate that the commercially available FCoV S gene mutation RT-qPCR evaluated in the present study (at least in its current form) was not able to correctly identify the S gene mutations previously described to be specific for FIP [11]. In view of these results, this assay in its current form cannot be recommended for the diagnosis of FIP. On the other hand, sequencing of the regions of interest within the FCoV S gene could potentially be more suitable for the detection of S gene mutations, since no mutated FCoV sequences were detected by Sanger sequencing.

One limitation of this study was the fact that collection of the samples occurred postmortem. Therefore, it is possible that the amount of viable viral RNA was degraded. Another limitation was that the material analyzed by RT-qPCR was not exactly the same as the material that was sequenced; the examined material originated from the same biopsies, but the RNA was extracted independently. In addition, the commercial RT-qPCR assays were performed from July 2018 to August 2019 and sequencing was done in April 2021; however, all samples had been stored at −20 °C until processing. Finally, another limitation of the study was that sequencing was not performed on all samples. Sequencing was performed to specifically verify the correctness of results of samples classified as “mutated FCoV” in FCoV S gene mutation RT-qPCR.

## 5. Conclusions

FCoV RNA can be detected in cats without FIP, not only in feces but also systemically in various tissues and body fluids, but none of the detected FCoV contained S gene mutations as confirmed by Sanger sequencing of the S gene region. The applied commercially available FCoV S gene mutation RT-qPCR, however, classified FCoV as “mutated FCoV” or “mixed FCoV” in several samples. The results of the present study thus indicate that the accuracy of the herein used commercial FCoV S gene mutation RT-qPCR is limited, and other methods, such as quantification of FCoV RNA loads, should be used for the diagnosis of FIP. Alternatively, or additionally, sequencing of the regions of interest within the S gene is recommended for detection of S gene mutations which can increase diagnostic specificity when compared to detection of FCoV by 7b gene RT-qPCR alone.

## Figures and Tables

**Figure 1 viruses-14-01671-f001:**
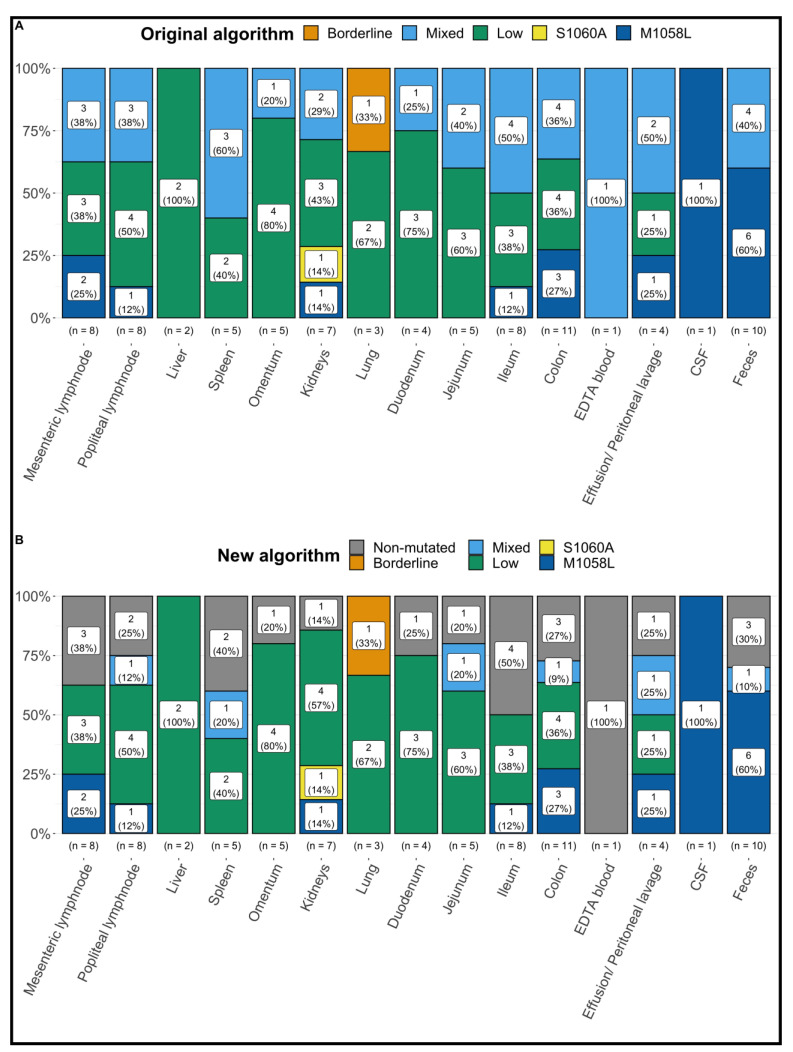
Summary of the results of the commercial real-time feline coronavirus (FCoV) 7b gene and spike (S) gene mutation reverse transcription polymerase chain reactions (RT-qPCR) following following the original algorithm (**A**) and the new algorithm (**B**) in all tissue, fluid, and fecal samples from the 21 cats positive for FCoV RNA in at least one sample. M1058L = positive FCoV S gene mutation RT-qPCR detecting mutation in nucleotide 23,531, corresponding to amino acid substitution M1058L; S1060A = positive FCoV S gene mutation RT-qPCR detecting mutation in nucleotide 23537, corresponding to amino acid substitution S1060A; Low = positive FCoV 7b gene RT-qPCR with viral load below cut-off (therefore no further differentiation possible by FCoV S gene mutation RT-qPCR); Mixed = positive FCoV 7b gene RT-qPCR and presence of FCoV with and without S gene mutations; Borderline = a specific PCR product (amplicon) could neither be detected with certainty nor could its presence be ruled out; EDTA = ethylenediamine tetra acetic acid; CSF = cerebrospinal fluid.

**Figure 2 viruses-14-01671-f002:**
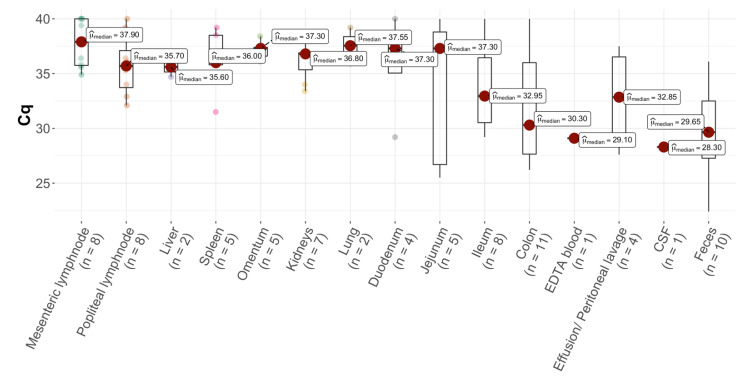
**Box-whisker-plot comparing** Cq values in tissue, fluid, and fecal samples from the 21 cats positive for feline coronavirus (FCoV) RNA in at least one sample. EDTA = ethylenediamine tetra acetic acid; CSF = cerebrospinal fluid.

**Table 1 viruses-14-01671-t001:** Results of the commercial real-time feline coronavirus (FCoV) 7b gene reverse transcription polymerase chain reaction (RT-qPCR) and spike (S) gene mutation RT-qPCR following the original algorithm, and the new algorithm for interpretation of the results in the 87 cats.

			Original Algorithm	New Algorithm
		**Total number of cats**	87	87
**FCoV 7b Gene RT-qPCR**		**FCoV 7b gene RT-qPCR-** negative in all samples	66	66
		**FCoV 7b gene RT-qPCR-** positive in at least 1 sample	21	21
**FCoV S Gene Mutation RT-qPCR^0^**	**Same Results in All Samples**	** ^1^ ** **Only Non-mutated**	0	0
		** ^2^ ** **Only Mutated (M1058L *)**	2	2
		** ^3^ ** **Only Mutated (S1060A ^¶^)**	0	0
		** ^4^ ** **Only Mixed ^§^**	1	0
		Only Low ^‡^	7	7
	**Different Results in Different Samples**	**Mixed ^§^** and Mutated (**M1058L ***)	1	0
		**Mixed ^§^** and Low ^‡^	4	0
		Borderline ^Ω^, **Mixed ^§^** and Low ^‡^	1	0
		**Mutated (M1058L *)** and Low ^‡^	3	3
		**Mutated** (both **M1058L *** and **S1060A ^¶^**) and Low ^‡^	1	1
		**Mixed ^§^** and **Mutated (M1058L *)** and Low ^‡^	1	0
		Non-mutated and Low ^‡^	0	4
		Non-mutated and **M1058L ***	0	1
		Non-mutated and **Mixed ^§^**	0	1
		Non-mutated, **M1058L *, Mixed ^§^** and Low ^‡^	0	1
		Non-mutated, Borderline ^Ω^, **Mixed ^§^** and Low ^‡^	0	1

M1058L * = positive FCoV S gene mutation RT-qPCR detecting mutation in nucleotide 23531, corresponding to amino acid substitution M1058L; S1060A ^¶^ = positive FCoV S gene mutation RT-qPCR detecting mutation in nucleotide 23537, corresponding to amino acid substitution S1060A; Low ^‡^ = positive FCoV 7b gene RT-qPCR with viral load below cut-off (therefore no further differentiation possible by FCoV S gene mutation RT-qPCR); Mixed FCoV ^§^ = positive FCoV 7b gene RT-qPCR and presence of FCoV with and without S gene mutations; Borderline ^Ω^ = a specific PCR product (amplicon) could neither be detected with certainty nor its presence could be ruled out; ^0^ FCoV S gene mutation RT-qPCR was only performed on the 21 FCoV 7b gene RT-qPCR-positive samples; ^1^ Only Non-mutated FCoV = non-mutated FCoV in each positive sample; ^2^ Only M1058L * = M1058L * in each positive sample; ^3^ Only S1060A ^¶^ = S1060A ^¶^ in each positive sample; ^4^ Only Mixed FCoV ^§^ = mixed FCoV ^§^ in each positive sample; Bold indicates samples positive for Mixed ^§^, M1058L * or S1060A ^¶^.

**Table 2 viruses-14-01671-t002:** Results of the commercial real-time feline coronavirus (FCoV) 7b gene and spike (S) gene mutation reverse transcription polymerase chain reactions (RT-qPCR) following the original algorithm and the new algorithm in all tissue, fluid, and fecal samples from the 21 cats positive for FCoV RNA in at least one sample.

Cat		Mesenteric Lymph Node	Popliteal Lymph Node	Liver	Spleen	Omentum	Kidneys	Lung	Duodenum	Jejunum	Ileum	Colon	EDTA Blood	Effusion/Peritoneal Lavage	CSF	Aqueous Humor	Feces
1	Cq	39.4							>40.0		39.0						
	Origi-nal	Low ^‡^	-	-	-	-	-	-	Low ^‡^	-	Low ^‡^	-	-	-	-	-	-
	New	Low ^‡^	-	-	-	-	-	-	Low ^‡^	-	Low ^‡^	-	-	-	-	-	-
2	Cq				31.5			39.2									
	Origi-nal	-	-	-	**Mixed** ^§^	-	-	Low ^‡^	-	-	-	-	-	-	-	-	-
	New				Non-mutated			Low ^‡^									
3	Cq	34.9	35.3	34.7	38.5		34.0										
	Origi-nal	**Mixed** ^§^	Low ^‡^	Low ^‡^	Low ^‡^	-	Low ^‡^	-	-	-	-	-	-	-	-	-	-
	New	Non-mutated	Low ^‡^	Low ^‡^	Low ^‡^		Low ^‡^										
4	Cq						38.3										
	Origi-nal	-	-	-	-	-	Low ^‡^	-	-	-	-	-	-	-	-	-	-
	New						Low ^‡^										
5	Cq	35.6	34.0	36.5	35.8	36.1	36.7	35.9	29.2	26.7	29.2	26.2	29.1	27.6			27.5
	Origi-nal	**Mixed ^§^**	**Mixed ^§^**	Low ^‡^	**Mixed ^§^**	**Mixed ^§^**	**Mixed ^§^**		**Mixed ^§^**	**Mixed ^§^**	**Mixed ^§^**	**Mixed ^§^**	**Mixed ^§^**	**Mixed ^§^**	-	-	**Mixed ^§^**
	New	Non-mutated	Non-mutated	Low ^‡^	Non-mutated	Non-mutated	Non-mutated	Low ^‡^	Non-mutated	Non-mutated	Non-mutated	Non-mutated	Non-mutated	Non-mutated			Non-mutated
6	Cq	>40.0	>40.0														
	Origi-nal	Low ^‡^	Low ^‡^	-	-	-	-	-	-	-	-	-	-	-	-	-	-
	New	Low ^‡^	Low ^‡^														
7	Cq										>40.0						
	Origi-nal	-	-	-	-	-	-	-	-	-	Low ^‡^	-	-	-	-	-	-
	New										Low ^‡^						
8	Cq											>40.0					
	Origi-nal	-	-	-	-	-	-	-	-	-	-	Low ^‡^	-	-	-	-	-
	New											Low ^‡^					
9	Cq					38.4											
	Origi-nal	-	-	-	-	Low ^‡^	-	-	-	-	-	-	-	-	-	-	-
	New					Low ^‡^											
10	Cq													29.5	28.3		
	Origi-nal	-	-	-	-	-	-	-	-	-	-	-	-	M1058L *	M1058L *	-	-
	New													M1058L *	M1058L *		
11	Cq	Cq 35.8	35.2 (FNA) 32.1			37.3	36.8			>40.0	30.7	28.1		36.2			
	Origi-nal	**M1058L ***	**M1058L ***	-	-	Low ^‡^	**M1058L ***	-	-	Low ^‡^	**M1058L ***	**M1058L ***	-	Low ^‡^	-	-	-
	New	**M1058L ***	**M1058L ***			Low ^‡^	**M1058L ***			Low ^‡^	**M1058L ***	**M1058L ***		Low ^‡^			
12	Cq									38.8							
	Origi-nal	-	-	-	-	-	-	-	-	Low ^‡^	-	-	-	-	-	-	-
	New									Low ^‡^							
13	Cq	>40.0	36.4		39.2	36.6	36.8	No Cq value	37.0	25.5	30.0	26.3					23.2
	Origi-nal	**Mixed ^§^**	**Mixed ^§^**	-	**Mixed ^§^**	Low ^‡^	**Mixed ^§^**	Borderline ^Ω^	Low ^‡^	**Mixed ^§^**	**Mixed ^§^**	**Mixed ^§^**	-	-	-	-	**Mixed ^§^**
	New	Non-mutated	Non-mutated		**Mixed ^§^**	Low ^‡^	Low ^‡^	Borderline ^Ω^	Low ^‡^	**Mixed ^§^**	Non-mutated	**Mixed ^§^**					**Mixed ^§^**
14	Cq	36.4	39.2		36.0	37.4	33.4				35.6	27.2					22.4
	Origi-nal	**M1058L ***	Low ^‡^	-	Low ^‡^	Low ^‡^	**S1060A ^¶^**	-	-	-	Low ^‡^	**M1058L ***	-	-	-	-	**M1058L ***
	New	**M1058L ***	Low ^‡^		Low ^‡^	Low ^‡^	**S1060A ^¶^**				Low ^‡^	**M1058L ***					**M1058L ***
15	Cq	>40.0										37.8					27.2
	Origi-nal	Low ^‡^	-	-	-	-	-	-	-	-	-	Low ^‡^		-	-	-	**M1058L ***
	New	Low ^‡^										Low ^‡^					**M1058L ***
16	Cq		32.9				37.0		37.6	37.3	33.1	29.6					32.2
	Origi-nal	-	**Mixed ^§^**	-	-	-	Low ^‡^	-	Low ^‡^	Low ^‡^	**Mixed ^§^**	**M1058L ***	-	-	-	-	M1058L *
	New		**Mixed ^§^**				Low ^‡^		Low ^‡^	Low ^‡^	Non-mutated	**M1058L ***					M1058L *
17	Cq										32.8	34.2					30.6
	Origi-nal	-	-	-	-	-	-	-	-	-	**Mixed ^§^**	Low ^‡^	-	-	-	-	Mixed ^§^
	New										Non-mutated	Low ^‡^					Non-mutated
18	Cq											33.8					32.6
	Origi-nal	-	-	-	-	-	-	-	-	-	-	Low ^‡^	-	-	-	-	M1058L *
	New											Low ^‡^					M1058L *
19	Cq											38.7					36.1
	Origi-nal	-	-	-	-	-	-	-	-	-	-	**Mixed ^§^**	-	-	-	-	M1058L *
	New											Non-mutated					M1058L *
20	Cq		36.1									30.3		37.5			28.7
	Origi-nal	-	Low ^‡^	-	-	-	-	-	-	-	-	**Mixed ^§^**	-	**Mixed ^§^**	-	-	Mixed ^§^
	New		Low ^‡^									Non-mutated		**Mixed ^§^**			Non-mutated
21	Cq value																34.2
	Origi-nal	-	-	-	-	-	-	-	-	-	-	-	-	-	-	-	**M1058L ***
	New																**M1058L ***

Original = Original algorithm for FCoV S gene mutation RT-qPCR; New = new algorithm for FCoV S gene mutation RT-qPCR; Cq = crossing point value of FCoV 7b gene RT-qPCR; M1058L* = positive FCoV S gene mutation RT-qPCR detecting mutation in nucleotide 23531, corresponding to amino acid substitution M1058L; S1060A ^¶^ = positive FCoV S gene mutation RT-qPCR detecting mutation in nucleotide 23537, corresponding to amino acid substitution S1060A; - = negative FCoV 7b gene RT-qPCR; Low ^‡^ = positive FCoV 7b gene RT-qPCR with viral load below cut-off (therefore no further differentiation possible by FCoV S gene mutation RT-qPCR); Mixed ^§^ = positive FCoV 7b gene RT-qPCR and presence of FCoV with and without S gene mutations; Borderline ^Ω^ = a specific PCR product (amplicon) could neither be detected with certainty nor could its presence be ruled out; EDTA = ethylenediamine tetra acetic acid; CSF = cerebrospinal fluid; Bold indicates samples positive for Mixed ^§^, M1058L *or S1060A.

**Table 3 viruses-14-01671-t003:** Results of the commercial real-time feline coronavirus (FCoV) spike (S) gene mutation reverse transcription polymerase chain reaction (RT-qPCR) and corresponding Sanger sequencing and cloning in the 16 samples (8 tissue and 8 fecal samples) of 9 nine cats that could be successfully sequenced.

Cat	Material	Commercial FCoV S Gene Mutation RT-qPCR According to the Original Algorithm	Commercial FCoV S Gene Mutation RT-qPCR According to the New Algorithm	Sanger Sequencing	Sequencing after Cloning
5	Feces	Mixed ^§^	Non-mutated	Non-mutated	Non-mutated
11	Popliteal lymph node	M1058L *	M1058L *	Non-mutated	Non-mutated
11	Kidney	M1058L *	M1058L *	Non-mutated	
11	Ileum	M1058L *	M1058L *	Non-mutated	Non-mutated
11	Mesenteric lymph node	M1058L *	M1058L *	Non-mutated	
13	Feces	Mixed ^§^	Mixed ^§^	Non-mutated	Non-mutated
14	Kidney	M1058L *	M1058L *	Non-mutated	Non-mutated
14	Colon	M1058L *	M1058L *	Non-mutated	Non-mutated
14	Feces	M1058L *	M1058L *	Non-mutated	Non-mutated
16	Ileum	Mixed ^§^	Non-mutated	Non-mutated	
16	Colon	M1058L *	M1058L *	Non-mutated	
16	Feces	M1058L *	M1058L *	Non-mutated	Non-mutated
17	Feces	Mixed ^§^	Non-mutated	Non-mutated	
18	Feces	M1058L *	M1058L *	Non-mutated	
19	Feces	M1058L *	M1058L *	Non-mutated	
20	Feces	Mixed ^§^	Non-mutated	Non-mutated	

M1058L * positive FCoV S gene mutation RT-qPCR detecting mutation in nucleotide 23531, corresponding to amino acid substitution M1058L; Mixed ^§^ = presence of both FCoV with and without S gene mutations.

## Data Availability

The authors confirm that the datasets analyzed during the study are available from the corresponding author upon reasonable request.

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
