# Peer review of "Detection of Feline Coronavirus Variants in Cats without Feline Infectious Peritonitis"

_viruses, 2022, doi:10.3390/v14081671_

Round 1

Reviewer 1 Report

In the introduction and throughout the paper the authors should make clear that they are talking about FCoV type I and not FCoV type II which can also cause FIP. It may seem trivial but there is some confusion in literature and it should always be clear to the reader about which FCoV the authors are talking. 

Line 73 to 77 is repetitive. 

The first 5 lines of discussion are also repetitive. 

Author Response

Reviewer: 1

Reviewers report for the author

viruses-1729100 review

In the introduction and throughout the paper the authors should make clear that they are talking about FCoV type I and not FCoV type II which can also cause FIP. It may seem trivial but there is some confusion in literature and it should always be clear to the reader about which FCoV the authors are talking. 

This is correct. We added this information.

Line 47-49: There are two serotypes of FCoV: type I and type II [6–8]. Serotype I is considered of only feline origin, while serotype II originates from a recombination of canine coronavirus and feline coronavirus [8,9].

Line 96: “… FCoV type I variants…”

Line 73 to 77 is repetitive. 

We thank the reviewer for the information. We made the respective changes.

Line 80-85: “Reverse transcription polymerase chain reaction (RT-PCR) is frequently applied to detect FCoV RNA in different samples, but FCoV RNA can also be detected in cats without FIP [15,19–22]. However, cats without FIP generally have significantly lower FCoV RNA loads measured by quantitative RT-PCR than cats with FIP [15], and thus,…”

The first 5 lines of discussion are also repetitive.

We agree. We made the respective changes.

Line 467-469: “The objective of the present study was to detect FCoV S gene variants in different tissue, body fluid, and fecal samples of cats in which FIP was excluded and that died from unrelated diseases. This is the first study that included a large number of different tissue, body fluid, and fecal samples from a defined population of cats in which FIP was definitively ruled out by gold standard methods. The results of the commercially available FCoV S gene mutation RT-qPCR were quite extraordinary and unexpected, and therefore confirmation by a second method, Sanger sequencing, was performed.”

Reviewer 2 Report

The authors of the manuscript entitled ‘Detection of feline coronavirus variants in cats without feline infectious peritonitis’ set out to determine the presence and genotype of FCoV in confirmed FIP negative cats, in multiple tissues and body fluids, using a combination of clinically relevant diagnostic RT-qPCR and Sanger sequencing. The work described contributes to the debate concerning the diagnostic value of commonly used RT-qPCR assays targeting specific mutations in the S gene of FCoV, and offers some evidence supporting the designation of the S gene M1058L substitution as specific to FIP pathogenesis, rather than as a feature of systemic dissemination of FCoV. The study includes a relatively large cohort of cats in which FIP was definitively excluded (using histopathology and IHC, the current ‘gold standard’ for diagnosis), and is strengthened by the dual approach to S gene variant detection.

One weakness of the study design is the lack of validation of either the RT-qPCR or RT-PCR/Sanger sequencing using positive controls, i.e. RNA isolated from different tissues of histopathologically/IHC confirmed FIP cases. Although application of these methods to such samples has been reported elsewhere, and is correctly cited by the authors, inclusion of these samples in the present study would assuage the discussed concerns of possible ‘methodological errors’ and would also strengthen the suggestion that the M1058L substitution in the FCoV S gene is in fact specific to FIP. It is noted that a reference is made to such results from one of the authors of this study in the Discussion, and another manuscript under review is cited, however without access to said manuscript it is this reviewer’s opinion that direct inclusion of these data in the present manuscript would have been beneficial.

Specific comments

Table 2 is difficult to read, with some inconsistencies:

·       For cat 13, two Cp values are reported for the FCoV 7b RT-qPCR. This does not appear to be explained in the text or the caption for Table 2.

·       In addition, the lung sample for cat 13 is recorded as having ‘No Cp value’ for the FCoV 7b RT-qPCR but has been included in the FCoV S gene mutation RT-qPCR as a FCoV positive sample.

The summary/total table at the bottom of Table 2 showing sample numbers in each group for each tissue is helpful, but could be presented as a summary plot/figure (e.g. stacked bar charts for each algorithm).  

A minor point is that ‘Cp’ is not commonly used, and Cq is widely regarded as the standard nomenclature when reporting results from RT-qPCR (see e.g. PMID: 19246619 and PMID: 19223324).

There are statements in both the Abstract (lines 26-27) and the Discussion (line numbers for this section are unavailable in the manuscript version received for review, but the statement is ‘sequencing of the regions of interest within the FCoV S gene seems to be more suitable for the detection of S gene mutations’, which is in the final sentence of the penultimate paragraph) which, to varying degrees, assert that Sanger sequencing is more suitable than the RT-qPCR assay used in this study, and clinically, to identify FCoV S gene mutations. This is not directly supported by the data presented in the manuscript, since no mutated FCoV sequences were detected by Sanger sequencing. The final sentence of the Abstract is therefore incorrect and should be removed. The aforementioned statement in the Discussion is more speculative in tone, but should be framed in the context of other similar studies which have included FIP positive cats and sequenced mutated FCoV from cats without FIP (e.g. PMID: 28982390). Although this article is referenced earlier, there is limited discussion of the discrepancy between studies.

Overall, the manuscript is presented clearly with a detailed summary of the current state of the field and relevant citations, and the methods used are described in sufficient detail. The tables used to display the data are appropriate and generally easy to understand. With the exceptions noted above, the conclusions drawn are consistent with the evidence and arguments presented.

Author Response

Reviewer: 2

Reviewers report for the author

viruses-1729100 review

Comments and Suggestions for Authors:

The authors of the manuscript entitled ‘Detection of feline coronavirus variants in cats without feline infectious peritonitis’ set out to determine the presence and genotype of FCoV in confirmed FIP negative cats, in multiple tissues and body fluids, using a combination of clinically relevant diagnostic RT-qPCR and Sanger sequencing. The work described contributes to the debate concerning the diagnostic value of commonly used RT-qPCR assays targeting specific mutations in the S gene of FCoV, and offers some evidence supporting the designation of the S gene M1058L substitution as specific to FIP pathogenesis, rather than as a feature of systemic dissemination of FCoV. The study includes a relatively large cohort of cats in which FIP was definitively excluded (using histopathology and IHC, the current ‘gold standard’ for diagnosis), and is strengthened by the dual approach to S gene variant detection.

One weakness of the study design is the lack of validation of either the RT-qPCR or RT-PCR/Sanger sequencing using positive controls, i.e. RNA isolated from different tissues of histopathologically/IHC confirmed FIP cases. Although application of these methods to such samples has been reported elsewhere, and is correctly cited by the authors, inclusion of these samples in the present study would assuage the discussed concerns of possible ‘methodological errors’ and would also strengthen the suggestion that the M1058L substitution in the FCoV S gene is in fact specific to FIP.

It is noted that a reference is made to such results from one of the authors of this study in the Discussion, and another manuscript under review is cited, however without access to said manuscript it is this reviewer’s opinion that direct inclusion of these data in the present manuscript would have been beneficial.

Concerning the positive controls for Sanger Sequencing:

We thank the reviewer 2 for addressing this point. As referenced in the manuscript, the method for the amplification of the region of the S-gene encompassing the M1058L mutation has been described in detail in a previously published study (Lutz et al. 2020). Furthermore, the method has been applied on effusion and blood samples originating from histopathologically confirmed FIP cases, where the mutated leucine at position 1058 was described, as reported in a study that has been recently accepted for publication (Meli et al. 2022). We have now included the accepted publication with the reference directly in the manuscript and hope that the statements are clearer now, as they can be found directly in the cited publication.

Line 311-314: “Furthermore, this method has been applied on effusion and blood samples originating from histopathologically confirmed FIP cases, where the mutated leucine at position 1058 was described, as reported by Meli et al. [36].”

Line 759-762: Meli, M. L.; Spiri, A. M.; Zwicklbauer, K.; Krentz, D.; Felten, S.; Bergmann, M.; Dorsch, R.; Matiasek, K.; Alberer, M.; Kolberg, L.; von Both, U.; Hartmann, K.; Hofmann-Lehmann, R., Fecal Feline Coronavirus RNA Shedding and Spike Gene Mutations in Cats with Feline Infectious Peritonitis Treated with GS-441524. Viruses 2022, 14, (5). 

Concerning the positive controls for RT-qPCR:

We agree with the reviewer and added this point.

Line 240-255:

RT-qPCR was run with six quality controls, including: (1) RT-qPCR-positive controls (using synthetic DNA covering the real-time PCR target region); (2) RT-qPCR-negative controls (PCR-grade nuclease-free water); (3) negative extraction controls (extraction positions filled with lysis solution and PCR-grade nuclease free water only); (4) RNA pre-analytical quality control targeting feline ssr rRNA (18S rRNA) gene complex; (5) a swab-based environmental contamination monitoring control; and (6) an internal positive control spiked into the lysis solution to monitor the nucleic acid extraction efficiency, and presence or absence of inhibitory substances (using lambda phage DNA). These controls assessed the functionality of the PCR test protocols (1), for the absence of contamination in the reagents (2) and laboratory (5), absence of cross-contamination during the extraction process (3), quality and integrity of the RNA as a measure of sample quality (4), and absence of PCR inhibitory substances as a carry-over from the sample matrix (6).

Table 2 is difficult to read, with some inconsistencies

For cat 13, two Cp values are reported for the FCoV 7b RT-qPCR. This does not appear to be explained in the text or the caption for Table 2.

We thank the reviewer for this comment. This was indeed a mistake. We corrected this in the new version so that there is only one CT value for cat 13 as well.

In addition, the lung sample for cat 13 is recorded as having ‘No Cp value’ for the FCoV 7b RT-qPCR but has been included in the FCoV S gene mutation RT-qPCR as a FCoV-positive sample.

We thank the reviewer for pointing this out. This sample was "borderline positive". For this reason, IDEXX did not provide us with the CT value of this sample. This is the only sample that was “borderline positive”. We added this classification accordingly in Materials and Methods and in Table 1 and 2.

Line 231-239: “4) “borderline” if a specific PCR product (amplicon) could neither be detected with certainty nor its presence could be ruled out, 5) “mutated FCoV” if the FCoV 7b gene RT-qPCR was positive and the mutated to non-mutated ratio was >2 (either mutation in nucleotide 23531 or 23537), or 6) “mixed FCoV”, if the FCoV 7b gene RT-qPCR was positive and the ratios of mutated to non-mutated fell in between “mutated FCoV” and “non-mutated FCoV” (1.5-2).”

Line 353: Table 1: for different results in different samples one row added with Old algorithm: Mixed, Low, Borderline: 1 and New algorithm: Non-mutated, Mixed, Low, Borderline: 1.

Line 371: Table 2, Cat 13: Lung: added “Borderline” instead of Low‡ for the old and new algorithm.

Line 359 and Line 418: added to legend of table 1 and table 2: "borderline”= if a specific PCR product (amplicon) could neither be detected with certainty nor its presence could be  ruled out.

The summary/total table at the bottom of Table 2 showing sample numbers in each group for each tissue is helpful, but could be presented as a summary plot/figure (e.g. stacked bar charts for each algorithm).  

We see the reviewer´s point. The changes were made as requested by the reviewer. We added 3 figures.

Line 432:

Figure 1. Summary of the results of the commercial real-time feline coronavirus (FCoV) 7b gene and spike (S) gene mutation reverse transcription polymerase chain reactions (RT-qPCR) following the original algorithm in all tissue, fluid and fecal samples from the 21 cats positive for FCoV RNA in at least one sample.

Line 438:

Figure 2. Summary of the results of the commercial real-time feline coronavirus (FCoV) 7b gene and spike (S) gene mutation reverse transcription polymerase chain reactions (RT-qPCR) following the new algorithm in all tissue, fluid and fecal samples from the 21 cats positive for FCoV RNA in at least one sample.

Line 440:

Figure 3. Cq values in all tissue, fluid and fecal samples from the 21 cats positive for FCoV RNA in at least one sample.

A minor point is that ‘Cp’ is not commonly used, and Cq is widely regarded as the standard nomenclature when reporting results from RT-qPCR (see e.g. PMID: 19246619 and PMID: 19223324).

We thank the reviewer for pointing this out. The word “Cp” was replaced by “Cq” throughout the paper (Line 312 (Table 2) and 412).

There are statements in both the Abstract (lines 26-27) and the Discussion (line numbers for this section are unavailable in the manuscript version received for review, but the statement is ‘sequencing of the regions of interest within the FCoV S gene seems to be more suitable for the detection of S gene mutations’, which is in the final sentence of the penultimate paragraph) which, to varying degrees, assert that Sanger sequencing is more suitable than the RT-qPCR assay used in this study, and clinically, to identify FCoV S gene mutations. This is not directly supported by the data presented in the manuscript, since no mutated FCoV sequences were detected by Sanger sequencing. The final sentence of the Abstract is therefore incorrect and should be removed.

We agree with the reviewer´s comments and removed the final sentence of the Abstract and this part in the discussion was revised.

Line 569-572: “On the other hand, sequencing of the regions of interest within the FCoV S gene is potentially more suitable for the detection of S gene mutations, although this was not directly shown in the present study since no mutated FCoV sequences were detected by Sanger sequencing.”

The aforementioned statement in the Discussion is more speculative in tone, but should be framed in the context of other similar studies which have included FIP positive cats and sequenced mutated FCoV from cats without FIP (e.g. PMID: 28982390). Although this article is referenced earlier, there is limited discussion of the discrepancy between studies.

We thank the reviewer for his comments and his valuable input. We added the information.

Line 538-548: According to other studies, the presence of M1058L or S1060A substitutions in the FCoV S protein were supposed to be a hallmark of systemic spread of FCoV rather than a definitive marker of the development of FIP [19,23]. In these studies, mutated FCoV was detected by sequencing in cats without FIP in tissue samples and in an effusion sample. One possible explanation for this discrepancy might be geographical difference in FCoV strains.

However, the results of the present study, in which mutated FCoV could not be detected in tissues from cats without FIP, together with the results of the study by Meli et al. [36] in which mutated FCoV was detected by sequencing in various tissues and body fluids of cats with FIP, support the hypothesis that the mutations are indeed a marker for FIP and not only for systemic spread of FCoV [19].

Round 2

Reviewer 1 Report

On line 174 you should include the citation or the primers used for control 4 targeting the feline ssrRNA. 

The caption for figures 1 and 2 is identical. It is not clear what is the difference between these graphs. If the graphs are comparing the two methods, perhaps these should be on a single figure so the reader can see it more clear. 

Author Response

On line 174 you should include the citation or the primers used for control 4 targeting the feline ssrRNA

We understand the reviewer´s point. However unfortunately, IDEXX, as a commercial laboratory, cannot not provide us with the information on the primers.

The caption for figures 1 and 2 is identical. It is not clear what is the difference between these graphs. If the graphs are comparing the two methods, perhaps these should be on a single figure so the reader can see it more clear.

We thank the reviewer for this comment. However, there is a slight but important difference in the title of the two figures. Figure 1 (now 1A) summarizes the results calculated with the old algorithm and figure 2 (now 1B) summarizes the results calculated with the new algorithm. This is included in the caption of the figures. We have adjusted the legend of the figures. We hope that the difference is clearer now.
